# The Fate of Transplanted Periodontal Ligament Stem Cells in Surgically Created Periodontal Defects in Rats

**DOI:** 10.3390/ijms20010192

**Published:** 2019-01-07

**Authors:** Kengo Iwasaki, Keiko Akazawa, Mizuki Nagata, Motohiro Komaki, Izumi Honda, Chikako Morioka, Naoki Yokoyama, Hirohito Ayame, Kazumasa Yamaki, Yuichi Tanaka, Tsuyoshi Kimura, Akio Kishida, Tetsuro Watabe, Ikuo Morita

**Affiliations:** 1Institute of Dental Research, Osaka Dental University, 8-1, Kuzuhahanazono-cho, Hirakata-shi, Osaka 573-1121, Japan; 2Department of Nanomedicine (DNP), Graduate School of Medical and Dental Sciences, Tokyo Medical and Dental University (TMDU), 1-5-45, Yushima, Bunkyo-ku, Tokyo 113-8510, Japan; 3Department of Periodontology, Graduate School of Medical and Dental Science, Tokyo Medical and Dental University (TMDU), 1-5-45, Yushima, Bunkyo-ku, Tokyo 113-8510, Japan; akakperi@tmd.ac.jp (K.A.); mizukin@umich.edu (M.N.); 4Kanagawa Dental University, Yokohama Clinic, 3-31-6, Tsuruya-cho, Kanagawa-ku, Yokohama, Kanagawa 221-0835, Japan; m.komaki@kdu.ac.jp; 5Department of Comprehensive Reproductive Medicine, Graduate School of Medical and Dental Science, Tokyo Medical and Dental University, 1-5-45 Yushima, Bunkyo-ku, Tokyo 113-8510, Japan; izumarch@gmail.com; 6Department of Pediatrics and Developmental Biology, Graduate School of Medical and Dental Science, Tokyo Medical and Dental University (TMDU), 1-5-45 Yushima, Bunkyo-ku, Tokyo 113-8510, Japan; cmorped@tmd.ac.jp; 7Biomaterial Laboratory, Research and Development Center, Dai Nippon Printing Co., Ltd., 250-1, Wakashiba, Kashiwa-city, Chiba 277-0871, Japan; yokoyama-n4@mail.dnp.co.jp (N.Y.); ayame-h@mail.dnp.co.jp (H.A.); Yamaki-K2@mail.dnp.co.jp (K.Y.); Tanaka-Y64@mail.dnp.co.jp (Y.T.); 8Institute of Biomaterials and Bioengineering, Tokyo Medical and Dental University (TMDU), 2-3-10, Kanda-Surugadai, Chiyoda-ku, Tokyo 101-0062, Japan; kimurat.mbme@tmd.ac.jp (T.K.); kishida.mbme@tmd.ac.jp (A.K.); 9Biochemistry, Graduate School of Medical and Dental Science, Tokyo Medical and Dental University (TMDU), 1-5-45, Yushima, Bunkyo-ku, Tokyo 113-8510, Japan; t-watabe@umin.ac.jp; 10Ochanomizu University, 2-1-1, Otsuka, Bunkyo-Ku, Tokyo 112-8610, Japan; morita.ikuo@ocha.ac.jp

**Keywords:** periodontal disease, regeneration, mesenchymal stem cells

## Abstract

Periodontal disease is chronic inflammation that leads to the destruction of tooth-supporting periodontal tissues. We devised a novel method (“cell transfer technology”) to transfer cells onto a scaffold surface and reported the potential of the technique for regenerative medicine. The aim of this study is to examine the efficacy of this technique in periodontal regeneration and the fate of transplanted cells. Human periodontal ligament stem cells (PDLSCs) were transferred to decellularized amniotic membrane and transplanted into periodontal defects in rats. Regeneration of tissues was examined by microcomputed tomography and histological observation. The fate of transplanted PDLSCs was traced using PKH26 and human Alu sequence detection by PCR. Imaging showed more bone in PDLSC-transplanted defects than those in control (amnion only). Histological examination confirmed the enhanced periodontal tissue formation in PDLSC defects. New formation of cementum, periodontal ligament, and bone were prominently observed in PDLSC defects. PKH26-labeled PDLSCs were found at limited areas in regenerated periodontal tissues. Human Alu sequence detection revealed that the level of Alu sequence was not increased, but rather decreased. This study describes a novel stem cell transplantation strategy for periodontal disease using the cell transfer technology and offers new insight for cell-based periodontal regeneration.

## 1. Introduction

Periodontal disease is characterized by the chronic inflammation of the tooth-supporting periodontal tissues, which is initiated by the infection of mainly gram-negative bacteria [1,2]. In the progressive form of the disease, extensive destruction of periodontal tissues is found and it leads to eventual loss of the affected tooth [3]. Periodontal disease is the biggest reason for tooth loss in developing countries nowadays. Many trials to regenerate lost periodontal tissues have been conducted and several bioactive molecules and bone substitutive materials have been applied clinically [4,5,6]. However, the indication and the amount of tissue regeneration by these treatments are limited and there is still a need for a new regenerative method for periodontal disease. Periodontal ligament (PDL) is the thin connective tissue located between the alveolar bone and the tooth root that bridges these two hard tissues [7]. It contains abundant collagen fibers and functions as a shock absorber to reduce the direct transmission of mastication force toward bone. Besides decreasing mechanical force, PDL has another important role in maintaining homeostasis and promoting wound healing of periodontal tissues by supplying stem/progenitor cells around periodontal tissues. PDL is thought to be a key tissue in periodontal regeneration [8,9]. Mesenchymal stem cells (MSCs) are cells originally isolated from bone marrow aspirate as plastic adherent colony-forming cells [10,11]. They are capable of differentiating into osteoblastic, adipogenic, and chondrogeinic cells under specific induction conditions in vitro and can form ectopic bone and cartilages in vivo, leading to the notion that they are novel stem cells in mesodermal tissues [12,13,14]. MSC-like cells have been isolated from PDL using a similar culture method for bone marrow MSCs (BM-MSCs) [15]. They are termed periodontal ligament stem cells (PDLSCs) and are thought to be putative adult stem cells in PDL [15,16]. While PDLSCs possess BM-MSC-like characteristics, such as trilineage differentiation capacity in vitro, cell-surface-specific antigen expression, and immunomodulatory and antiapoptosis function, it has been reported that PDLSCs have unique characteristics [17,18]. PDLSCs form cementum and a PDL-like, tooth-specific anatomical structure when they are transplanted into animals with hydroxyapatite particles, while BM-MSCs produce bone and bone-marrow-like structure [15,18]. These results indicate that PDLSCs are a unique cell population that has greater potential to form periodontal tissues and is the best cell source for periodontal regeneration.

We have recently invented a novel tissue engineering method, which we termed “cell transfer technology”, to transfer cultured cells onto scaffold materials by applying photolithography, which is often utilized in the printing and precision machinery industry [19,20]. One of the unique characteristics of this method is controllable cell transfer in the desired patterning on a scaffold [21]. However, in case of cell transplantation that requires cell number and not patterning, we can also transfer cells in a sheet format on a scaffold surface. We have demonstrated the regeneration of bone and periodontal tissues in vivo by transplanting cells in cell sheet format made using the cell transfer technology [22,23]. We used decellularized amniotic membrane (amnion) as a cell transfer scaffold, because of its flexibility and cell transfer efficiency. We have previously observed the new bone formation in the periodontal defect model in a rat maxilla fist molar, which mimicked class two furcation periodontal defect, after four weeks of transplantation of PDLSC-transferred amnion, and suggested the therapeutic potential of PDLSC-transferred amnion (PDLSC-amnion) [23]. However, in that study, this class of periodontal defect is relatively rare clinically. The examination in other common defect types is needed to broaden the range of application of the PDLSC-amnion approach. Additionally, the mechanisms of periodontal regeneration by PDLSC-amnion are not understood and the fate of transplanted PDLSCs in periodontal defects after transplantation has not been investigated.

Based on this background, we created a dehiscent-type periodontal defect in the rat mandibular first molar and transplanted PDLSC-amnion to investigate its therapeutic potential. We also examined the location of the transplanted PDLSCs and compared human genomic DNA in rat tissue sections obtained at different time points as healing proceeded, in order to trace the contribution of transplanted PDLSCs to the periodontal regeneration.

## 2. Results

### 2.1. Transplantation of PDLSC-Amnion

First, we created PDLSC-amnion by transferring cultured monolayers of PDLSCs onto the amnion surface using the novel cell transfer technology. Figure 1A demonstrates the schematic diagram of PDLSC-amnion fabrication. We made the tetraethylene glycol (TEG) or polyethylene glycol (PEG) layer on glass substrate. This TEG/PEG layer is hydrophobic and cells hardly adhere on this surface. The irradiation of UV on the TEG/PEG surface resulted in the partial degradation of TEG/PEG layer and the surface became hydrophilic. It was possible to control the hydrophobicity/hydrophilicity of the surface by changing the duration of UV irradiation. After the seeding of PDLSCs on the transfer substrate, PDLSCs adhered onto it and then we started transferring PDLSCs by placing the substrate in order to make direct attachment of the cell surface to the amnion. Here, PDLSCs were located between amnion and transfer substrate, adhering both amnion and TEG/PEG surface. Upon the removal of transfer substrate, PDLSCs were successfully transferred to the amnion when the binding between cells and amnion were stronger than that between cells and transfer substrate. We used cell transfer substrates that were optimized for transfer of PDLSCs in the pilot studies. Figure 1B shows the PDLSC-amnion created with GFP-transfected PDLSCs trimmed into squares. Next, we prepared periodontal defects on the buccal area of the rat first mandibular, according to the previous reports [24,25,26]. CT images and photographs of the buccal area of the mandible around first molar before and after the defect creation are shown in Figure 1C–F. We removed alveolar bone, PDL, cementum, and dentin to create defects that each measured 2 mm × 3 mm. After the defect formation, PDLSC-amnion or amnion was trimmed to the defect size and placed to cover the defect (Figure 1G). The PDLSC-amnion was oriented such that the cell surface directly attached to the denuded root surface.

### 2.2. Comparison of Periodontal Regeneration Using Micro-CT

For the quantification of regenerated tissues, histological quantification is common in animal experiments. However, the rat periodontal tissues were too small to obtain the same cross-sections to examine for reliable data collection. For this reason, we selected micro-CT data for the quantification of the regenerated tissue. Figure 2A shows the micro-CT images taken at four weeks after the transplantation. New bone formation was observed in micro-CT images from both control and PDLSC-amnion groups. The bone formation was always found at the bottom of the periodontal defect and the exposed root surface area was reduced at four weeks. Comparison of the exposed root areas revealed that more alveolar bone covered the root surface in the PDLSC-amnion group than in control group defects. Figure 2B shows the results of quantification of exposed root surface area at four weeks on micro-CT images. The exposed surface area was less in the PDLSC-amnion than in the control, indicating the enhanced periodontal regeneration in defects treated with PDLSC-amnion.

### 2.3. Histological Observations of Periodontal Defects

Figure 3 displays histological images of sections from periodontal defects four weeks after the transplantation. Sections made from the long axis (Figure 3A–H) and horizontal (Figure 3I–N) sectioning are shown.

Although new bone formation was observed in both control and PDLSC-amnion groups, the alveolar bone crest was found at a higher level in PDLSC-amnion section than in control (Figure 3A–D). In both groups, newly formed bone always accompanied the formation of narrow connective tissues space, which resembled PDL, and new tissues exhibited the normal periodontal tissue structure. Highly magnified images from the center of a defect showed that while most of the fibers ran parallel to the root surface in control sections, a cementum-like thin layer of hard tissue on the denuded dentin surface was evident in the PDLSC-amnion group (Figure 3E,F). In sections stained with azan, the fibers oriented parallel to the root surface were prominent in the control group, while in the PDLSC-amnion group, the fibers were embedded into a thin layer on the root surface and exhibited the structure of “Sharpey’s fibers” (Figure 3G,H).

In horizontal sections, the denuded root surface was covered with newly formed PDL-like tissue and bone in PDLSC-amnion sections, whereas the periodontal defect was mainly filled with dense bundles of fibers in control sections (Figure 3I–L). In azan-stained sections, fibers running parallel to the root surface were well characterized in control sections (Figure 3M). On the other hand, a PDL-like space was formed on the denuded root surface and fiber bundles and well-developed blood vessels were evident in PDLSC-amnion sections (Figure 3N). New formation of thin cementum-like tissue was also observed and perpendicularly oriented fibers were visible on the denuded root surface of the PDLSC-amnion group.

### 2.4. Localization of Transplanted Cells in Periodontal Defects

To further investigate the regenerative mechanism by PDLSC transplantation, we examined the localization of transplanted PDLSCs by transplanting cells labeled with PKH26. In fluorescence images of sections from PDLSC-amnion transplanted defects at four weeks (Figure 4A–E), the transplanted PDLSCs were found in several areas in the periodontal defect (Figure 4A,B). PKH26-positive cells were detected at the outer surface of regenerated alveolar bone (Figure 4A–C) and a small number of cells were scattered and found in the PDL space (Figure 4D). However, the vast majority of regenerated periodontal tissues comprised PKH26-negative cells (Figure 4E). We present the results of the cell trace experiment using three more rats as Appendix A. In these results, transplanted PDLSCs were also found in several limited areas in periodontal tissues four weeks post-transplantation. Sections from one rat were nearly free of transplanted cells.

### 2.5. Detection of Human-Derived Cells in Tissue Sections

Because PKH26 labeling is diluted by successive cell divisions, we next examined the presence of human cells by testing human DNA in rat sections. We isolated genome DNA samples from sections of periodontal defects obtained at 3, 7, and 28 days after cell transplantation and amplified the rat-specific housekeeping gene for GAPDH and the human-specific satellite gene Alu sequence. Figure 5 shows the results of gel running of PCR product samples for rat GAPDH and human Alu. The density of the rat GAPDH band remained almost throughout the sample collection period, while the intensity of the human Alu sequence was slightly decreased at day 28. These results suggested that the number of transplanted PDLSCs decreased with the time of sample collection.

## 3. Discussion

In the present study, we observed enhanced periodontal tissue formation by the transplantation of PDLSC-amnion. Consistent with our results, other studies have reported periodontal regeneration by cell transplantation [27,28,29]. It was originally anticipated that stem cell transplantation would lead to new tissue formation by the engraftment, growth, and differentiation of the transplanted stem cells. However, so far, the fate of transplanted cells in periodontal defects and regenerated tissues has been unclear. Three studies previously reported the fate of transplanted MSC in periodontal defects using labeling of cells. They reported that transplanted MSC were differentiated in osteoblasts, cementoblasts, and fibroblasts in regenerated periodontal tissues, suggesting the direct contribution of transplanted cells in new tissues [30,31,32]. These studies provided evidence that some transplanted MSCs can differentiate into new tissue-forming cells. However, the gross localization of transplanted cells in whole regenerated tissues was not determined. Recently, Yu et al. (2015) reported that GFP-labeled PDLSCs were detected around regenerated periodontal tissues three weeks after transplantation, with only a few cells detected as tissue-forming cells [33]. Their results are in line with our observation of only a few labeled PDLSCs in regenerated periodontal tissues. Moreover, human Alu sequence PCR results suggested the transplanted PDLSCs did not increase in number in periodontal tissues. These results indicate that the transplanted cells induced tissue regeneration, but that the majority of the new tissues were not formed by the transplanted cells themselves, indicating an indirect effect by the transplanted cells. Similar results showing the decrease of transplanted cells in spite of functional improvement have been reported in several disease models, including myocardial infarction, spinal cord injury, and multiple sclerosis [34,35,36,37].

In general, the microenvironment of recipient sites is a harsh place for transplanted cells to survive, because of inflammation in the wound-healing process and the continual ischemia due to the loss of blood vessels. Thus, it is conceivable that many transplanted PDLSCs did not survive in this microenvironment in the present study. The mechanism whereby PDLSCs induced periodontal regeneration when their number was decreased is unclear. One possible explanation is the indirect effect of MSCs to enhance wound healing through secreted factors. It has been demonstrated that PDLSCs secrete paracrine factors with various functions including antiapoptosis, immunomodulation, and anti-inflammation [17,38,39,40], and that these secreted factors in cells that survive in the transplanted site may positively function in periodontal wound healing. We recently reported that the transplantation of conditioned medium from PDLSCs could induce periodontal regeneration using the same rat periodontal defect model [24]. In that study, we found that the level of tumor necrosis factor-alpha was decreased in regenerating periodontal tissues where PDLSC-conditioned medium was transplanted, suggesting an anti-inflammatory function of the conditioned medium. We have also investigated the composition of the paracrine factors using LC/MS/MS and protein array analysis and found that PDLSC-conditioned medium contained various proteins including extracellular matrix proteins (type I collagen, fibronectin, etc.), growth factors (IGFBP6, PDGF-AA, etc.), and angiogenic factors (VEGF, uPA, etc.) [24]. We also observed that the condition medium from PDLSCs inhibited IFN-γ-induced *TNF-α* gene expression in mouse macrophage cell line, indicating the anti-inflammatory effect of PDLSC paracrine factors. These proteins may function as a promoter of periodontal regeneration in this study. Moreover, it has recently been demonstrated that many functions of MSC paracrine factors are mediated by extracellular vesicles, such as exosomes, which are secreted by the transplanted MSC [41,42]. Exosomes are the extracellular vesicles that have a double-layered lipid membrane and enclose many cellular functional components such as mRNA, miRNA, DNA, and proteins. After being released by cells, exosomes are incorporated by other distant cells and they then affect the recipient cell functions. Thus, exosomes are being considered as a new form of communication between cells. Our data suggested the PDLSCs paracrine effect to be a regenerative mechanism. Thus, it is possible that exosomes from PDLSCs might work as paracrine factors in periodontal regeneration. Because the PDLSC-conditioned medium contains exosomes, it is conceivable that exosomes may contribute to the regeneration of tissues in the conditioned medium and PDLSC-amnion. Further detailed studies are needed to clarify the involvement of exosomes in periodontal regeneration with PDLSC transplantation.

There are some limitations to the interpretation of our results. Firstly, our periodontal defects were surgically created and not induced by chronic inflammation as is the case in clinical settings. Bacterial infection and chronic inflammation may affect the results of regenerative therapy. Secondly, the age of the animals and the donors of PDLSCs were younger than the majority of periodontal patients. Thirdly, the observation time period was limited in this study; data on cell fate and periodontal regeneration at shorter and longer time points are needed for a more detailed understanding of periodontal regeneration by PDLSCs transplantation.

Our results demonstrated that PDLSCs decreased rather than increased after transplantation, and the regeneration of periodontal tissues was enhanced. For stem cell transplantation, the ex vivo expansion of cells is indispensable, but the phenotypic alteration of cells, including canceration, is an unavoidable issue. Although more conclusive research is needed, we demonstrated that the enhanced tissue regeneration is accompanied by the reduction of transplanted cells and this notion may reduce the safety issues.

We used the decellularized amnion as scaffold material for PDLSC transplantation. In our pilot study, we did not find any difference in the healing of periodontal tissues between amnion transplantation and the empty defect group. Because of the flexible and soft characteristics of amnion, it could not maintain space for new tissues to regenerate. Thus, the retention of PDLSCs in periodontal defects may be the most important function of the amnion in this study.

Various methodologies have been used and reported for the transplantation of PDLSCs in periodontal defects. They include transplantation of cells with scaffold materials, such as gelfoam [43,44] and hydroxyapatite/tricalcium phosphate [45,46], and in a cell sheet format created using a temperature-responsible culture plate [47]. When compared with these existing materials and methods, PDLSC transplantation using our cell transfer technology possesses two unique characteristics. First, we can place the layered PDLSCs to make direct contact with the denuded root surface by taking advantage of the flexible nature of our material. PDLSCs have the capacity to differentiate into cementoblasts, and our method enables the transplantation of the potential cementoblast progenitors in a way that mimics the anatomical position of cementoblasts. This is an advantage of cell sheet format transplantation that cannot be achieved by any other cell transplantation method such as gel and solid-bone-substituted materials. The second unique characteristic is the easy and reliable handling of the cell transplantation material during transplantation manipulations that is possible because of the cell transfer technology. Periodontal defects often display a narrow and complicated geometrical shape that makes it difficult to transplant fragile materials, because of the limited accessibility. Thus, the deformation of the cell sheet and disturbance of cells in the cell sheet are unavoidable during transplantation. We have reported that cultured cells firmly adhered to the amnion surface after using our novel cell transfer method, and that cells are not removed from the membrane after deformations including stretching, holding, and dragging [20,23]. Our method makes it possible to transplant cell sheets easily and reliably to the defect, because it is possible to drag and stretch the membrane extensively using surgical instruments. We transplanted PDLSC-amnion to periodontal defects, taking full advantage of our method to cover the denuded root surface. The cell transplantation method demonstrated in this study has many advantages for cell transplantation in periodontal defects.

In this study, we compared the periodontal regeneration between amnion alone and amnion plus PDLSC groups. Because scaffold material is crucial for cell retention, and the aim of this study was to investigate the periodontal regeneration, after amnion + PDLSC transplantation, we did not establish a cell-alone group in this study. We previously demonstrated that amnion + osteoblast induced more bone formation than only osteoblast injections in a mouse calvarial bone defect model and suggested the importance of the amnion as a scaffold material in cell-based regenerative treatment [23].

## 4. Materials and Methods

### 4.1. Cell Culture

The protocol of this study was approved by the ethics committee for clinical research at Tokyo Medical and Dental University (D2013-021, 18 March 2014). PDL tissues were removed from premolars and third molars extracted from healthy subjects 14–28 years of age. The tissues were minced using a surgical knife (Feather Safety Razor, Osaka, Japan) for 15 min in 500 μL of digestion solution (α-MEM (Invitrogen, Carlsbad, CA, USA) with 3 mg/mL of collagenase type I (WAKO Pure Chemicals, Osaka, Japan,) and 4 mg/mL of dispase (Invitrogen). Then, 4.5 mL of digestion solution was poured onto the minced PDL solution and incubated in a water bath at 37 °C for 60 min with continuous agitation. The reaction solution was centrifuged and 10 mL αMEM containing 15% fetal bovine serum (High Clone, Logan, UT, USA) was added to make a cell suspension. To remove debris, the cell suspension was passed through a cell strainer with a pore size of 70 µm (BD Falcon, Bedford, MA, USA). The culture medium was changed every 3 days. Colonies of PDLSCs were harvested and passaged. As demonstrated by our previous study [20,23,24], PDLSCs exhibited unique cell surface antigen expression profiles such as CD105+, CD90+, CD44, CD146+, CD73+, CD31−, CD34−, and CD45−, as well as the ability to differentiate into steoblasts, adipocytes, and chondrobytes. To minimize the characteristic differences among each batch of cells in this study, we used the cells at passage numbers below 7. In some experiments, green fluorescence protein (GFP)-expressing PDLSCs (PDLSC-GFP) were established as previously described [22] and PDLSCs were labeled with PKH 26 fluorescent cell linker (Sigma-Aldrich, St. Louis, MO, USA) according to the protocol provided from the manufacture.

### 4.2. Amniotic Membrane Preparation

The research protocol was approved by the ethics committee for clinical research at Tokyo Medical and Dental University (M2000-1102, 01 January 2000). Human fetal membranes were obtained from patients undergoing cesarean delivery at Tokyo Medical and Dental University Hospital. The fetal membrane was washed with sterile saline and cut into small pieces. The pieces were incubated for 1 h at 37 °C in 0.02% EDTA-PBS. The chorion was removed mechanically from the amnion using a cell scraper. To remove cell components from the amnion, the membranes were decellularized using a high hydrostatic pressure treatment as reported previously [48]. The amnion was stored in Dulbecco’s modified Eagle’s medium containing 50% glycerol (Invitrogen) at −80 °C until use.

### 4.3. PDLSCs Transfer onto Amnion

Cultured PDLSCs were transferred onto the amnion as previously reported [20,23] and summarized in Figure 1A. Briefly, degradation of tetraethylene glycol (TEG) or polyethylene glycol (PEG) layer on glass substrate was induced with ultraviolet (UV) irradiation. PDLSCs were seeded onto the hydrophilic surface of transfer substrate (5 × 10^5^ cells/cm^2^) and incubated for 3 h. Then, the transfer substrate was placed downward onto the amnion surface. After overnight incubation, the transfer substrate was gently removed using sterile surgical forceps.

### 4.4. Transplantation of PDLSC-Transferred Amniotic Membrane in Periodontal Defects

All study protocols and procedures were approved by the Animal Care Ethics Committee of Tokyo Medical and Dental University. The periodontal defects were surgically created as described previously with minor modification [24,25,26]. Briefly, a total of 22 (*n* = 10 regeneration, *n* = 4 cell labeling, *n* = 6 human Alu PCR, *n* = 2 dead during surgery) male athymic nude rats (F344/N-Jcl-rnu/rnu; age 7–8 weeks) were anesthetized using isoflurane (Abbott Laboratories, Queenborough, UK) and pentobarbital sodium (Kyoritsu Seiyaku, Tokyo, Japan). An extraoral incision was made at the bottom of the mandible and the buccal plate was exposed (Figure 1E). The buccal bone, PDL, cementum, and dentin from the mesial root of the mandibular first molar to the mesial root of the second molar were carefully removed to create a periodontal defect using rotatory instruments (Figure 1F). Each periodontal defect in the buccal area was 2 mm in height and 3 mm in width. After creation of the defect, PDLSC-amnion (experimental) or amnion (control) was trimmed and placed to cover the periodontal defect and the masseter and skin were sutured with 7-0 or 5-0 silk (Mani, Tochigi, Japan). The decision of treatment in each experiment was made randomly by a third person blinded to the treatment allocation after the periodontal defect had been created.

### 4.5. Histological Analysis

Four weeks after transplantation, the rats were sacrificed with an overdose of isofulrane and the mandible block was removed. The block was fixed in 4% paraformaldehyde-phosphate buffer for 2 days and decalcified in 10% EDTA at 4 °C for 4 weeks, followed by dehydration, paraffin embedding, and sectioning. The sections were observed microscopically using a BZ800 apparatus (Keyence, Osaka, Japan) after staining with hematoxylin-eosin and azan. For labeled cell observation, the section was mounted with 4′,6-diamidino-2-phenylindole (DAPI) mounting solution (Vector Laboratories Inc., Burlingame, CA, USA).

### 4.6. Microcomputed Tomography (Micro-CT) Analysis

We used micro-CT for the comparison of regenerated bone between test groups, because of its reliable quantification of hard tissues. Four weeks after transplantation, the healing of periodontal defects was evaluated using micro-CT scans with a model SMX-100CT apparatus (Shimadzu Co., Kyoto, Japan). Three-dimensional image construction was performed using VG Studio MAX 2.0 (Volume Graphics, Heidelberg, Germany). To examine the periodontal tissue healing, we measured the exposed root surface area up to the line connecting the mesial and distal points of the cement enamel junction (CEJ) of the mandibular first molar using BZ-analyzer software (Keyence, Osaka, Japan).

### 4.7. Detection of Human Genome from Rat Sections

Nine sections of paraffin-embedded mandible cut in longitudinal sections were acquired from rats sacrificed at 3, 7, and 28 days after PDLSC-amnion transplantation. Genomic DNA was isolated from the samples using NucleoSpin FFPE DNA (Macherey-Nagel, Düren, Germany). Rat glyceraldehyde-3-phosphate dehydrogenase (GAPDH) and human Alu genomic sequences were amplified using rat GAPDH primer (forward: GGGAGAGCTGGGTTTGTTTT, reverse: TGGAAGAATGGGAGTTGCTG, product size = 204 bp) and human Alu primer (forward: GGCGCGGTGGCTCACG, reverse: TTTTTTGAGACGGAGTCTCGCTC, product size = 282 bp). PCR products were analyzed by 2% agarose gel electrophoresis.

### 4.8. Statistical Analysis

Statistical differences of the experimental groups were determined using Student’s *t* test. Differences with *p* < 0.05 were considered significant.

## 5. Conclusions

In the present study, we demonstrate that periodontal regeneration is induced by transplanting PDLSCs using a novel tissue engineering method, cell transfer technology, suggesting the value of this method as a new cell-based treatment for periodontal disease. A limited number of transplanted PDLSCs were detected in periodontal defects after four weeks, and the drastic engraftment of the cells was not found. As far as we know, this is the first report to quantitatively examine the fate of transplanted cells in periodontal defects. This study provides a new cell transplantation strategy for periodontal disease and insight regarding the mechanism of periodontal regeneration. Our data suggest a new scenario of periodontal regeneration by PDLSC transplantation, other than the engraftment, growth, and differentiation of the transplanted cells.

## Figures and Tables

**Figure 1 ijms-20-00192-f001:**
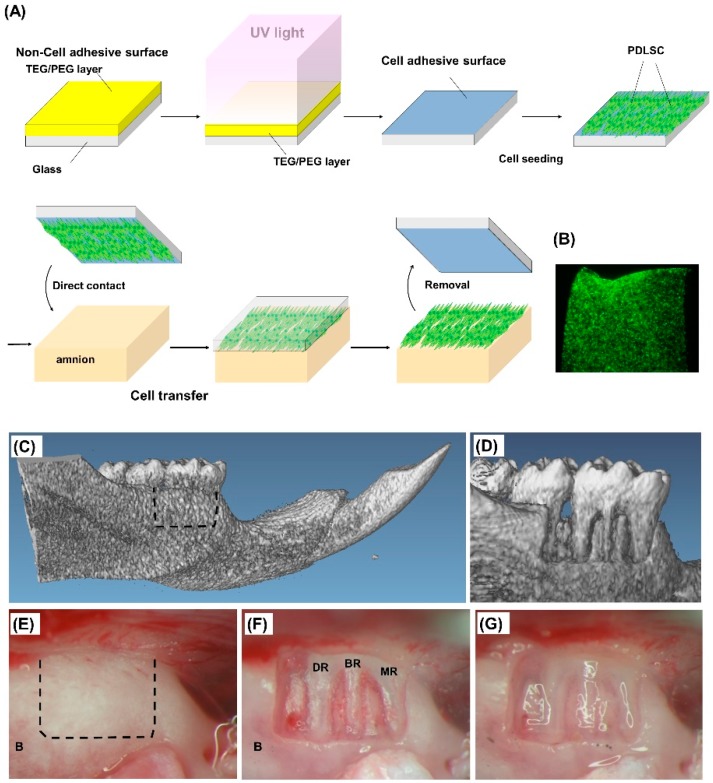
PDLSC transfer on amnion using the cell transfer technology and transplantation into periodontal defects. (**A**) Cultured PDLSCs were transferred onto decellularized amnion surface. The yellow area on the transfer base shows the hydrophobic cell nonadhesive surface and the blue area denotes the hydrophilic cell adhesive surface. Trypsinized PDLSCs were seeded onto blue cell adhesive surface, created by ultra violet light irradiation of the tetraethylene glycol (TEG) or polyethylene glycol (PEG) layer. After incubation for 3 h, PDSLCs adherent to the transfer substrate were placed onto amnion with the cell surface oriented downward. After overnight incubation, the transfer base was removed with forceps and PDLSCs were transferred from the transfer base to the amnion surface. (**B**) Transfer of PDLSC-GFP onto the amnion. GFP-transfected PDLSCs were transferred onto the amnion using the cell transfer technology, and then cut into squares using a surgical knife. PDLSCs were observed in cell sheet format. (**C**) CT image of the buccal side of a rat mandible. The dotted line outlines the periodontal defect. (**D**) CT image of a periodontal defect. By removing bone, PDL, cementum, and dentine, a 2 × 3 mm periodontal defect was created. (**E**) Photograph of a bone surface before periodontal defect creation. The dotted line outlines the periodontal defect. (**F**) Photograph of a periodontal defect. The defect was created from the mesial root of first mandibular to the mesial root of second molar. Exposure of the mesial, buccal, and distal roots of the first molar is evident in the defect. (**G**) Photograph of a periodontal defect after PDLSC-amnion placement. PDLSC-amnion was trimmed to cover the defect with a surgical knife and placed onto the periodontal defect with the cell surface oriented to attach to the denuded root.

**Figure 2 ijms-20-00192-f002:**
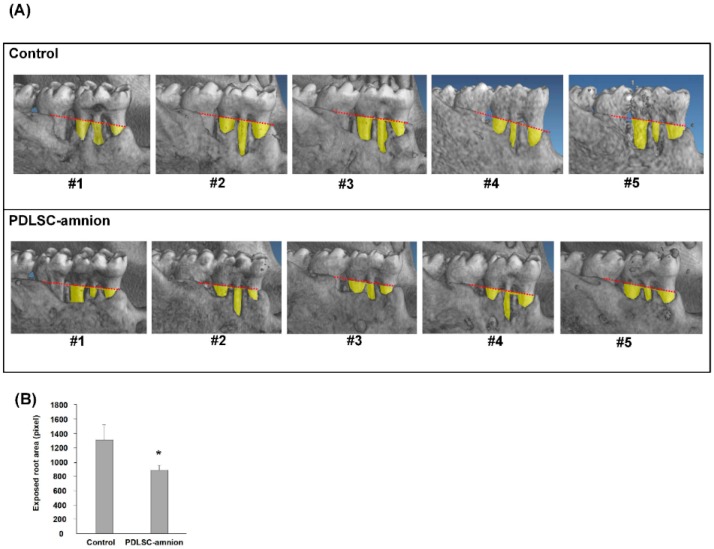
Periodontal regeneration in rat periodontal defects. (**A**) Micro-CT images of periodontal defects four weeks post-transplantation. The buccal area of the rat first mandibular molar was demonstrated from five rats in each test group. The yellow area shows the denuded root surface area below the red line, connecting the mesial and distal alveolar bone crests. (**B**) Quantification of the yellow area in micro-CT images taken at four weeks after transplantation. The yellow area is smaller in the PDLSC-amnion group than in the control, suggesting the enhanced new tissue formation. * *p* < 0.05, Student’s *t* test.

**Figure 3 ijms-20-00192-f003:**
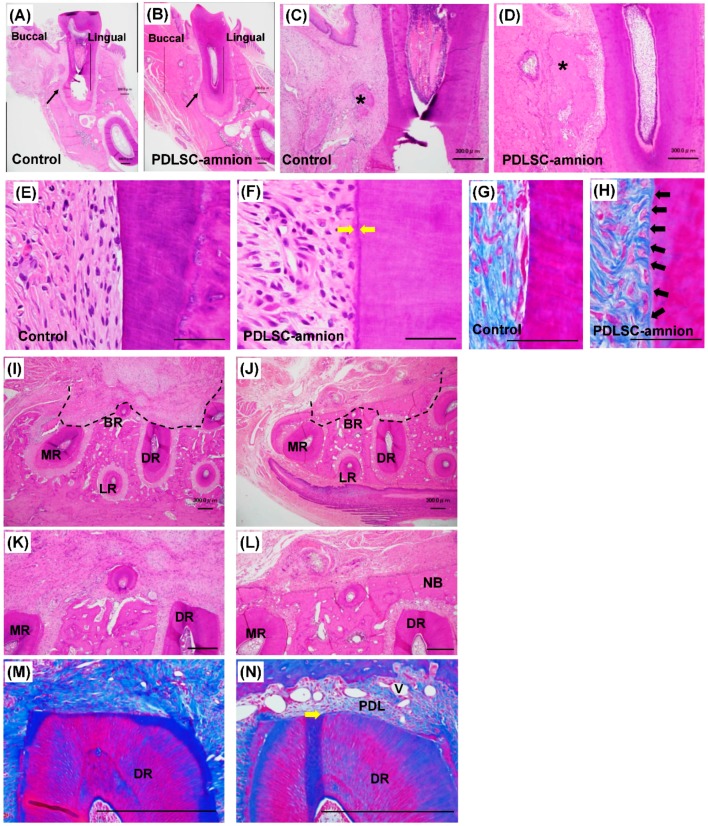
Histological analysis of periodontal tissues. Histological observation of periodontal tissues in rat periodontal defects four weeks after transplantation. Histological images are shown using long axial (**A**–**H**) and horizontal (**I**–**N**) sections at lower (**A**,**B**,**I**,**J**) and higher magnification (**C**,**D**,**E**–**H**,**K**–**N**). Sections were stained with hematoxylin and eosin (**A**–**F**,**I**–**L**) or azan (**G**,**H**,**M**,**N**). The boxed area in panels **A**,**B** are demonstrated in panels **C**,**D**, respectively. Close-up images of center of denuded root surface are shown in panels **E**–**H**. More bone formation was observed in the PDLSC-amnion group compared with control (**A**–**D**,**I**–**L**), and cementum-like thin layer of hard tissue was formed on the root surface in the PDLSC-amnion group (**F**,**N**, yellow arrows). Collagen fibers originating perpendicular to the root surface were obvious in PDLSC-amnion group sections (**H**, black arrows). The dotted line outlines the periodontal defect in horizontal sections (**I**,**J**). New formation of bone and PDL-like structure were prominent in the PDLSC-amnion group, while the defects were filled with collagen fibers in the control (**K**,**M**). The PDL space contained many blood vessels (**L**,**N**). *: new bone, black arrowhead: bottom of defect, yellow arrow: cementum-like tissue, black arrow: collagen bundles run from root surface, MR: mesial root, BR: buccal root, DR: distal root, LR: lingual root, NB: new bone, PDL: periodontal ligament, V: blood vessel, Bar = 300 μm (**A**–**D**,**I**–**N**) and 50 μm (**E**–**H**).

**Figure 4 ijms-20-00192-f004:**
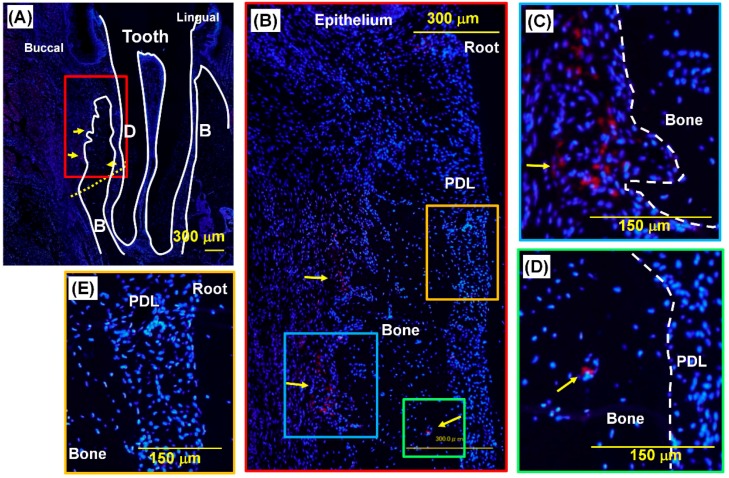
Localization of PKH26-labeled PDLSC in regenerated periodontal tissues. Fluorescence microscopic images of periodontal tissues four weeks after transplantation. Lower magnification image of the section (**A**). Close-up image of the red box area in panel (**A**)is shown in panel (**B**). Higher magnified images of the blue, green, and blue boxed areas are demonstrated in panels (**C**–**E**), respectively. The yellow and white dotted lines denote the bottom of created periodontal defect and bone surface, respectively. Red fluorescence signals are derived from PKH26, which was used for labeling of transplanted PDLSCs, and blue signals are from DAPI nuclear staining. PKH2-positive cells were observed in limited areas in periodontal tissues, as indicated by the yellow arrows, at the outer surface of bone and incorporated in blood vessel. The center area of regenerated PDL did not contain PKH26-positive cells. Yellow arrow: PKH26-positive cells, B: bone, D: dentin, PDL: periodontal ligament.

**Figure 5 ijms-20-00192-f005:**
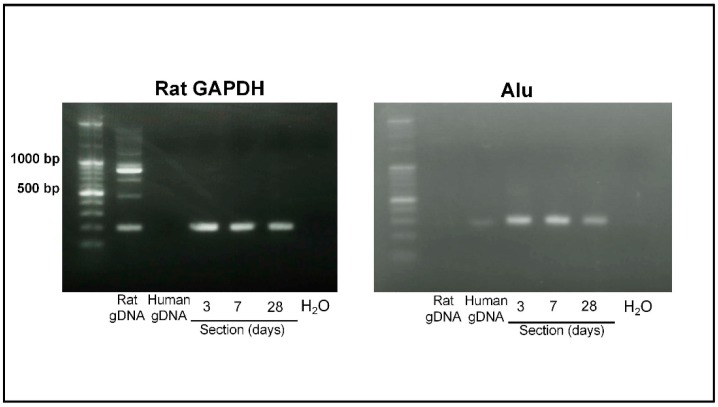
Electrophoresis of PCR products for rat GAPDH and human Alu sequence. PCR products of rat GAPDH and human Alu were analyzed by 2% agarose gel electrophoresis with 100 bp DNA marker and photographed. DNA samples from sections acquired at day 3, 7, and 28 showed similar intensity of rat GAPDH bands (204 bp), while the intensity of the human Alu sequence (282 bp) was slightly decreased. gDNA: genomic DNA.

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
