# Peer review of "The Fate of Transplanted Periodontal Ligament Stem Cells in Surgically Created Periodontal Defects in Rats"

_ijms, 2019, doi:10.3390/ijms20010192_

Reviewer 1 Report

The paper is well designed and the experiments is well performed.

Author Response

Thank you for your comment. We are grateful for your comment that our manuscript is acceptable to the International Journal of Molecular Science.

Reviewer 2 Report

This paper describes the creation of periodontal defects in rats for evaluating the effects of transplanted human periodontal ligament stem cells (PDLSCs) on the regeneration of periodontal tissues.  Transplantation of the PDLSCs was carried out as a sheet platform by using cell transfer technology including amniotic membrane.  Obtained results and conclusions were persuasive.  The quality of this paper fulfills the requirements of publication in International Journal of Molecular Sciences.  The reviewer recommends that the manuscript should be revised with the following comments:

 Major comments

1.     The authors did not provide the mechanism of the cell transfer technology throughout the manuscript.  For example, in Figure 1A, non-cell adhesive surface was converted into cell adhesive surface by UV irradiation.  The reviewer cannot understand why the surface should be converted, and whether the converted surface was suitable for transferring a cell sheet onto the amniotic membrane.  The authors should explain the details about the cell transfer technology.

2.     The authors did not mention the role of amniotic membrane during periodontal tissue regeneration.  The authors should explain whether decellularized amniotic membrane affect the regeneration of periodontal tissues or not.

 Minor comments

1.     In line 270, the authors indicated that PDLSC-conditioned medium contained some ECMs and growth factors.  The authors should indicate the reference or evidence of them.

2.     There are some grammatical errors.  For example, “Fort” in line 135, “Compared of” in line 141, and sentences in lines 292 and 301. 

3.     In line 133, “Bone regeneration by micro-CT” is inappropriate.  The reviewer recommended that the subheading should be rewritten.

4.     In the caption of Figure 4, there are no explanations about fluorescent colors.  The author should indicate what blue and red colors were.

Author Response

1. Mechanism of cell transfer technology

Thank you for your precious comment regarding the mechanisms of cell transfer technology. I agree with the reviewer that the information about the mechanisms of cell transfer technology was lacking in the original manuscript.

In the procedures of this method, we make the tetraethylene glycol(TEG) or polyethylene glycol(PEG) layer on glass plate. This TEG/PEG layer is hydrophobic and cells can not adhere on this surface. The irradiation of UV on the TEG/PEG surface results in the partial degradation of TEG/PEG layer and the surface becomes hydrophilic. We can control the hydrophobicity/hydrophilicity of the surface by changing the duration of UV irradiation. After the seeding of PDLSCs on the transfer substrate, PDLSCs adhere onto it and then we start transferring PDLSCs by placing the substrate in order to make direct attachment of the cell surface to amnion. Here, PDLSCs are located between amnion and transfer substrate, adhering both amnion and TEG/PEG surface. Upon the removal of transfer substrate, PDLSCs are successfully transferred to the amnion when the binding between cells and amnion are stronger than that between cells and transfer substrate. We used cell transfer substrates that optimally adjusted for transfer of PDLSCs in the pilot studies.

This detailed explanation of mechanisms of cell transfer technology was lacking in the original manuscript. Responding the precious input from the reviewer, above mentioned information were added in the revised manuscript. (Line 104 to 114)

Additionally, we used TEG and PEG for cell transfer substrate surface. We added PEG, which was missing in original manuscript in Figure 1 and the Results section. (Line 128, 383-384, Figure 1A.)

We appreciate the reviewer’s important comment.

2. The role of amnion in periodontal regeneration in this study

Thank you for your important query regarding the role of amnion in periodontal regeneration. In the pilot studies, we observed no enhancement of periodontal regeneration by amnion transplantation. Although amnion has been reported to improve the results of wound healing in some in vivo models, we did not find any difference in the healing of periodontal tissues between amnion transplantation and empty defect group. Because of the flexible and soft characteristics of amnion, the membrane could not maintain the space for tissues to regenerate. Thus, the retention of PDLSCs in periodontal defect maybe the most important function of the amnion in this study. Responding the reviewer’s comment, we added some sentences discussing the role on amnion in the Discussion section. (Line 313-317)

Minor points

1. The contents of PDLSC-conditioned medium

We appreciate the important comment about the protein contents of PDLSC-conditioned medium. We apologize that the important reference about PDLSC-conditioned medium was missing in original manuscript. The reference is Ref #24, which was published in Tissue engineering part A in 2017 from our laboratory. In that study, we examined the protein contents of PDLSC-conditioned medium using liquid chromatography–tandem mass spectrometry (LC/MS/MS)(shown in Supplementary data in Ref#24) and several protein arrays (Angiogenesis array, Growth factor array and Cytokine array shown in Figure 5 in Ref#24). Results from LC/MS/MS demonstrated that extracellular matrix proteins including type I collagen and fibronectin were the top peptides detected in PDLSC-conditioned medium.

According to the comments from the reviewer, we corrected the description and added the reference in the Discussion section. (Line 282-285)

2. Grammatical mistakes

Thank you for pointing out the grammatical mistakes in the original manuscript. We corrected the indicated grammatical mistakes and checked again throughout the revised manuscript. (Line 145, 151-153, 302-304, 310-312)

3. Mistake in the description “Bone regeneration by micro-CT”

Thank you for the indication. We changed the description to make correct subheading in the revised manuscript. (Line 143)

4. Information of fluorescence in caption of Figure 4

Thank you for the query. We added the information regarding red and blue fluorescence in the Figure 4 caption in the revised manuscript. (Line 236-237)

Reviewer 3 Report

The article “The fate of transplanted periodontal ligament stem cells in surgically created periodontal defects in rats” by Kengo Iwasaki et al. is dedicated to an experimental evaluation of cell transfer technology to transfer cells onto a scaffold surface and assessment the potential of such technique for regenerative medicine. Authors utilized common precision machinery and additive manufacturing method – photolithography in a new capacity – to transfer cultured cells onto scaffold materials, and study the viability of the transferred cells. The experimental results presented by authors are important, interesting, and method is novel.  The manuscript is of great quality, with extensive details on experimental methods and techniques used to acquire the data. In my opinion the article should be published in MDPI International Journal of Molecular Sciences, as is, without revisions.

Author Response

Thank you for your comment about our manuscript. We are grateful for your comment that our manuscript is acceptable to the International Journal of Molecular Science.